# Peripheral Blood Biomarkers Predictive of Efficacy Outcome and Immune-Related Adverse Events in Advanced Gastrointestinal Cancers Treated with Checkpoint Inhibitors

**DOI:** 10.3390/cancers14153736

**Published:** 2022-07-31

**Authors:** Zhening Zhang, Tong Xie, Changsong Qi, Xiaotian Zhang, Lin Shen, Zhi Peng

**Affiliations:** Key Laboratory of Carcinogenesis and Translational Research, Department of Gastrointestinal Oncology, Peking University Cancer Hospital & Institute, Beijing 100142, China; shenshengzhiguang@163.com (Z.Z.); xietong1995@126.com (T.X.); xiwangpku@126.com (C.Q.); zhangxiaotianmed@163.com (X.Z.); linshenpku@163.com (L.S.)

**Keywords:** gastrointestinal cancer, peripheral blood biomarkers, immune checkpoint inhibitors, neutrophil-to-lymphocyte ratio, immune-related adverse events

## Abstract

**Simple Summary:**

Although immune checkpoint inhibitors improve the survival of patients with advanced gastrointestinal cancers, they also cause a series of immune-related adverse events, which could sometimes be lethal and may hamper the effectiveness of anticancer therapies. The purpose of this study was to explore clinically accessible biomarkers to predict survival and adverse events in patients with advanced gastrointestinal cancers treated with checkpoint inhibitors. In a retrospective cohort containing 243 patients, we found that early treatment lines, the presence of immune-related adverse events, and a lower posttreatment neutrophil-to-lymphocyte ratio were independent factors predicting superior prognosis. Good physical strength and a low posttreatment neutrophil-to-lymphocyte ratio were independent risk factors for immune-related adverse events. These findings may assist in identifying patients who are more likely to respond to immunotherapy and suffer from fewer toxicities, which is of great value in guiding clinical decisions.

**Abstract:**

Background: Gastrointestinal cancers constitute a major burden of global cancer mortalities. In recent years, the advent of immune checkpoint inhibitors has greatly improved the survival of patients with advanced gastrointestinal cancers, while predictive biomarkers of treatment efficacy and toxicities are still unmet demands. Methods: In our retrospective study, patients with advanced gastrointestinal cancers who received single or double immune checkpoint inhibitors in the Department of Gastrointestinal Oncology in Peking University Cancer Hospital between July 2016 and February 2022 were enrolled. Records of clinicopathological information, survival parameters, safety data, and baseline and posttreatment peripheral blood constituents were retrieved. Cox regression analysis and logistic regression analysis were performed to identify the predictive factors of treatment outcomes and immune-related adverse events. Results: We demonstrated that early treatment lines, the presence of immune-related adverse events, and a lower C2 neutrophil-to-lymphocyte ratio were independent factors predicting a superior objective response rate and progression-free survival in patients treated with immunotherapy. Lower ECOG PS, higher baseline albumin, and lower C2 neutrophil-to-lymphocyte ratios were independent risk factors for the onset of immune-related adverse events. Patients who succumbed to immune-related adverse events during immunotherapy presented better survival. Conclusion: Our results indicate that peripheral blood markers have potential for predicting treatment outcomes and immune-related adverse events in patients with advanced gastrointestinal cancer. Prospective validations are warranted.

## 1. Introduction

Gastrointestinal cancers (including gastric, esophageal, and colorectal cancer) accounted for 19.0% of cancer-related deaths worldwide in 2020 [1]. Despite the improvement of survival owing to the development of chemotherapy and targeted therapy, the prognosis of advanced gastrointestinal cancers remains poor [2,3,4]. Immune checkpoint inhibitors (ICIs), such as antibodies targeting programmed cell death-1 (PD-1), programmed cell death ligand-1 (PD-L1), and cytotoxic T-lymphocyte antigen-4 (CTLA-4), have revolutionized the treatment paradigm of gastrointestinal malignancies in recent years. Encouragingly, compared to standard chemotherapy, pembrolizumab monotherapy results in a significantly longer progression-free survival (PFS) and a higher objective response rate (ORR) in patients with microsatellite instability-high (MSI-H) colorectal cancers, as suggested by the Keynote-177 trial. The successes of the Keynote-180 and the Keynote-059 trials also promote the clinical use of ICIs in heavily treated refractory esophagus cancer and gastric cancer [5,6,7]. Medical oncologists are paying unremitting efforts to seek out the determinants of treatment responses and toxicities [8,9].

Conventional biomarkers, including PD-L1, microsatellite status, and tumor mutational burden (TMB), are insufficient and sometimes incompetent in predicting the therapeutic outcomes of ICIs [10,11,12,13,14]. Beyond that, a spectrum of unique adverse events, known as immune-related adverse events (irAEs), induced by overactivation of immune reactions toward normal tissues might cause severe treatment complications and limit the clinical application of ICIs [15,16,17]. IrAEs are generally diverse in their manifestations and might affect nearly all organ systems of the body. Patients could present a rapid attack within a few days or display a delayed onset several months after ICIs are initiated. Most irAEs are mild and manageable, while a minority of irAEs, such as pneumonitis and myocarditis, could be fatal if not identified in a timely manner [17,18,19]. Regarding interventions for severe irAEs, patients are required to withdraw ICIs permanently and receive long-term steroids or immunosuppressive agents [16,18,20,21]. Moreover, there is currently a lack of biomarkers capable of predicting the occurrence and severity of irAEs. 

It has been reported that circulating inflammatory cell components can reflect the magnitude of systemic inflammation, which plays an intricate role in the regulation of antitumor immunity [22,23,24,25,26]. According to previous literature, peripheral blood parameters, such as the neutrophil-to-lymphocyte ratio (NLR), platelet-to-lymphocyte ratio (PLR), and lymphocyte-to-monocyte ratio (LMR), could efficiently predict the responses to ICIs in multiple malignancies [22,26,27,28,29,30,31,32,33,34,35,36,37,38]. These indicators are both economical and accessible. In addition, albumin and lactate dehydrogenase (LDH) are parameters derived from routine blood biochemistry, which are linked with metabolic homeostasis and affect the therapeutic outcomes of immunotherapy [33,34,39,40]. Interestingly, the development of irAEs is strongly correlated with favorable prognosis in patients treated with ICIs. Thus, we speculate that irAEs and antitumor immune responses mediated by ICIs might share mutual mechanisms. Peripheral blood parameters could putatively serve as predictive factors for irAEs.

The aim of this study was to investigate the value of baseline peripheral blood biomarkers for predicting treatment outcomes and irAEs among patients with gastrointestinal cancers treated with ICIs. Our results could facilitate the precise identification of patients who benefit from immunotherapy while exampting irAEs, which provides great reference value in clinical practice.

## 2. Methods

### 2.1. Study Design and Participants

In this single-center, retrospective cohort study, patients aged between 18 and 85 with a pathological diagnosis of stage IV esophageal, gastric, or colon cancer who were treated in the Department of Gastrointestinal Oncology at Peking University Cancer Hospital between July 2016 and February 2022 were enrolled for analysis. Each eligible individual received at least one dose of either anti-PD-1, anti-PD-L1, or anti-PD-1 plus anti-CTLA-4 monoclonal antibodies. Patients who received other antitumor therapies concurrently, such as chemotherapy, targeted therapy, or radiotherapy, were excluded from the study. Patients diagnosed with preexisting autoimmune diseases were also excluded. The study was approved by the Medical Ethics Committee of Peking University Cancer Hospital. Written informed consent was exempted, and the authors declare that the study was conducted in accordance with the Helsinki Declaration.

### 2.2. Data Collection

Complete blood cell counts (neutrophils, lymphocytes, monocytes, platelets, etc.), LDH values, and albumin values from peripheral blood at baseline (within 7 days before the administration of immunotherapy, defined as C1) and at the second course (2 to 3 weeks after the first dose, defined as C2) were extracted from the electronic medical records. General demographic, disease-related, and treatment-related information (e.g., treatment type and treatment line) was also retrieved. The NLR is calculated from the absolute neutrophil count divided by the absolute lymphocyte count. The PLR is calculated from the absolute platelet count divided by the absolute lymphocyte count. The LMR is calculated from the absolute lymphocyte count divided by the absolute monocyte count. The cutoff values for NLR, PLR, and LMR as dichotomous variables are round-off numbers referenced by previous literature [22,27,38]. Due to the lack of uniform standards, we defined PD-L1 positivity in this study as ≥1% of the tumor and stromal cells being positive by immunohistochemistry staining. The microsatellite status of tumors was detected by PCR.

### 2.3. Definitions of Treatment Outcomes and Adverse Events

Treatment responses were assessed by investigators in accordance with the Response Evaluation Criteria in Solid Tumors (RECIST) version 1.1 through computed tomography (CT) scan. In each assessment, either a complete response or partial response was classified as an objective response; otherwise, there was no response. The ORR referred to the ratio of patients who achieved an objective response from the initiation of immunotherapy to the date of disease progression, death, or final follow-up. PFS was calculated from the date of initial treatment to disease progression or death. The Common Terminology Criteria for Adverse Events (CTCAE) of the National Cancer Institute (version 5.0) was used to determine patients’ adverse events. The irAEs were typically referred to as a range of adverse events that reflected a disorder of the immune system. In our study, any drug-related toxicity except infusion reaction that occurred within one year of treatment initiation was viewed as an irAE. The routine safety follow-up time was one year, unless patients died during this period. Incidences, categories, and grades of irAEs were documented according to CTCAE.

### 2.4. Statistical Analysis

Categorical variables were analyzed using Chi-square tests or Fisher exact tests. Continuous variables were analyzed using the Wilcoxon rank-sum test. PFS curves were calculated using the Kaplan–Meier method, and the two-sided log-rank test was used to evaluate differences. Cox regression models were exploited to determine the risk factors for PFS. Factors that were statistically significant in the univariate analysis were incorporated into the multivariate analysis. Logistic regression analysis was applied to identify risk factors for irAEs. GraphPad Prism 7.0 (GraphPad Software, La Jolla, CA, USA) and SPSS 25.0 (SPSS Software, Chicago, IL, USA) were used for statistical analyses. A two-sided *p* value < 0.05 was considered statistically significant.

## 3. Results

### 3.1. Patient Characteristics

Overall, 243 eligible patients were included for analysis (Table 1). Most of the patients had an ECOG PS of 0 to 1 (97.5%). The median age was 58 years (range 18 to 85), and a preponderance of men was indicated (70.8%). Patients with a confirmative diagnosis of advanced esophagus cancer, gastric cancer, or colon cancer accounted for 20.6%, 47.3%, and 32.1% of the total populations, respectively. All esophagus cancers had microsatellite stable (MSS) status, and 34 out of 115 gastric cancers and 73 out of 78 colon cancers had MSI-H status. More than half of the patients received anti-PD-1 therapies (58.8%), while the others received anti-PD-L1 or anti-PD-1 plus anti-CTLA-4 therapies (namely combinational immunotherapy). Among all participants, 28.8% were administered immunotherapy in the first-line setting.

### 3.2. Summary of irAEs

Immune-related adverse events were reported in 139 (57.2%) patients, with rash (20.6%), thyroiditis (17.3%), and transaminitis (13.2%) being the most common (Table 2). For organ-related toxicities, dermatological toxicities were the most commonly reported irAEs, followed by hepatologic and endocrinal toxicities. A total of 260 events of all grade irAEs were documented, of which 33 events (12.7%) were grade 3 or higher in severity. The most common grade 3 or higher irAEs were myositis (2.7%), transaminitis (2.3%), and rash (2.3%). The profile of irAEs classified by different treatment types was presented in Appendix A. Overall, the incidence rates of all-grade irAEs in the anti-PD-1 + CTLA-4, anti-PD-1, and anti-PD-L1 groups were significantly different (73.3%, 54.5%, and 57.5%, respectively, *p* = 0.044), while the incidence rates of grade ≥ 3 irAEs across these three groups were nonsignificant (18.3%, 10.5%, and 17.5%, respectively, *p* = 0.273). The majority of patients with grade 3 or higher irAEs were treated with steroids (84.8%) and discontinued immunotherapy permanently, while the others discontinued immune medications. All patients who received immunosuppressive therapies achieved alleviation of their irAEs.

### 3.3. Univariate and Multivariate Analyses of Risk Factors for ORR and PFS

For univariate analysis, esophageal-gastric cancer (*p* = 0.003), early treatment line (*p* < 0.001), PD-L1 positive expression (*p* = 0.044), presence of any grade of irAE (*p* < 0.001), presence of grade 3 to 4 irAE (*p* < 0.001), lower baseline LDH (*p* = 0.021), lower C2 NLR (*p* < 0.001), and lower C2 PLR (*p* = 0.006) were associated with higher objective response rate (Table 3). Esophageal-gastric cancer (*p* < 0.001), early treatment line (*p* < 0.001), presence of any grade of irAE (*p* < 0.001), presence of grade 3 to 4 irAE (*p* = 0.026), lower baseline LDH (*p* = 0.040), lower C1 NLR (*p* = 0.002), lower C2 NLR (*p* < 0.001), lower C2 PLR (*p* = 0.005), and lower C2 LMR (*p* < 0.001) were associated with longer PFS.

For multivariate analysis, early treatment line, presence of any grade of irAE, and lower C2 NLR remained significantly associated with higher ORR and PFS. In addition, female sex was another independent factor of improved ORR (*p* = 0.037) (Figure 1). Age ≥ 60 years (*p* = 0.014), colon cancer (*p* < 0.001), and lower baseline LDH (*p* = 0.016) were also independent factors of prolonged PFS. Subgroup analyses according to different tumor types were suggested in Appendix A.

### 3.4. Univariate and Multivariate Analyses of Risk Factors for irAEs

For univariate analysis, lower ECOG PS (*p* = 0.007), esophageal-gastric cancer (*p* = 0.031), combinational immunotherapy (*p* = 0.008), higher baseline albumin (*p* = 0.017), and lower C2 NLR (*p* = 0.013) were associated with an increased incidence rate of any grade of irAEs (Table 4). Risk factors for the incidence of severe irAEs (grade 3 to 4 irAEs) were not found.

For multivariable analysis, lower ECOG PS (*p* = 0.040), higher baseline albumin (*p* = 0.049), and lower C2 NLR (*p* = 0.044) remained risk factors for any grade of irAEs (Figure 2). Subgroup analyses according to different tumor types, treatment types, and microsatellite statuses are suggested in Appendix A.

### 3.5. Correlations between irAEs and Treatment Outcomes

Patients who developed irAEs had a significantly longer median PFS (14.8 versus 2.3 months, HR = 0.47 with *p* < 0.0001) than those who did not. However, the PFS of patients with irAEs did not differ significantly by irAE grade. In addition, according to the chi-square analysis, the ORR was significantly higher among patients with irAEs than among those without irAEs (78.4% versus 43.2%, *p* < 0.0001) (Figure 3). Subgroup analyses according to different tumor types, treatment types, and microsatellite statuses were suggested in Appendix A.

## 4. Discussion

Our retrospective study indicated that early treatment lines, the presence of irAEs, and a lower C2 NLR were independent factors predicting superior ORR and PFS in patients treated with immunotherapy. Lower ECOG PS, higher baseline albumin, and lower C2 NLR were independent risk factors for the development of irAEs. Patients who succumbed to irAEs during immunotherapy presented longer PFS and higher ORR.

The advent of immunotherapy has tremendously reshaped the therapeutics of gastrointestinal tumors, which improves the survival of patients with late-stage tumors to a certain extent. Despite this, the immune activation mediated by checkpoint inhibitors could be a two-edged blade, as responses and toxicities are interconnected and sometimes overlap in clinical scenarios [16,41]. IrAEs represent a group of drug toxicities disparate from adverse events caused by conventional chemotherapy, which leads to morbidity, debilitates long-term outcomes, and poses practical challenges for oncologists. In this context, early identification of potential beneficiaries and patients vulnerable to irAEs might avoid unnecessary financial cost, provide warning and monitoring for high-risk patients, and facilitate better management of treatment toxicities.

In recent years, many efforts have been made to determine the underlying mechanisms and biomarkers of treatment responses as well as toxicities regarding immunotherapy. Widely validated hallmarks of response in gastrointestinal tumors include PD-L1 expression, tumor mutation burden, and microsatellite state. In addition, a vast number of fundamental and translational studies have also uncovered other molecular features that are tightly associated with treatment responses and irAEs. For example, tumors typified by certain genetic alterations (e.g., POLE or POLD mutations), transcriptomic profiles (e.g., IFN-γ-related RNA signatures), tumor microenvironment constituents (e.g., enrichment of tumor infiltrating lymphocytes), and microbiota (e.g., EBV infection, gut flora diversity) exhibit higher sensitivity to immunotherapy [16,42,43,44,45,46]. However, the complicated interactions across these factors make any one of them not a competent predictor of responses. Moreover, despite the cumulative insights into the determinants of toxicities brought by immunotherapy (including but not limited to genetic predisposition, dysbiosis of microbiota, preexisting autoantibodies, and proinflammatory cytokines), ideal parameters for the prediction of irAEs are less well-defined [47,48,49,50,51,52,53,54]. In the clinical setting, efficient, convenient, and economical predictive biomarkers of irAEs are still warranted.

The NLR, PLR, and LMR are hematological indexes derived from common blood tests, which are both highly accessible and economical. Previous research has adequately demonstrated the prognostic value of baseline NLR and PLR in patients treated with anti-PD-1 antibodies [22,26,27,28,29,30,31,32,33,34,35,36,37]. Elevated baseline NLR and PLR have been reported to be associated with worse treatment outcomes and poor prognosis in multiple malignancies. In particular, for patients diagnosed with advanced melanoma and treated with nivolumab, those with a baseline NLR ≥ 5 had significantly worse overall survival and PFS than those with a baseline NLR < 5 [26]. Additionally, high pretreatment NLR and PLR together could efficiently predict poor survival of non-small cell lung cancer (NSCLC) patients treated with nivolumab and are independent of other known prognostic factors such as tobacco use and ECOG PS [31]. However, the utility of NLR and PLR in predicting the development of irAEs is controversial, although the responses to immunotherapy are coupled with irAEs in gastrointestinal cancers. According to a few observations, a lower baseline NLR is correlated with a higher incidence rate of irAEs, while in other reports, an elevated NLR is implicated in the occurrence of irAEs (especially interstitial pneumonitis) [55,56,57,58,59]. It should be noted that irAEs are a group of heterogeneous manifestations whose spectrum varies by cancer entity and treatment type. Thus, we assumed that different irAEs might have different pathogenic mechanisms and could exert distinct influences on peripheral blood components. In our study, posttreatment (but not pretreatment) NLR was an independent factor of therapeutic outcomes and irAEs. We attributed this finding to the fact that baseline hemocyte counts are more likely to be affected by confounding factors such as nonspecific inflammation or myelosuppression caused by previous cytotoxic medications. In contrast, posttreatment hematological indexes are much steadier, since the dynamic changes in neutrophils or lymphocytes along with immunotherapy are usually consistent, as proposed by previous literature [60,61].

The irAE frequency in our study was 56.3% (139/243) overall, 13.6% (33/243) for grade 3 to 4 events, which is generally consistent with the overall frequency of 30.7 to 57.9% for any grade, and 9.2 to 12.4% for grade 3 to 4 irAEs reported in the Keynote-177 and Keynote-180 clinical trials [6,7]. We noticed that the spectrum of irAEs in gastrointestinal tumors is distinct. As expected, rashes and thyroiditis were the two most common irAEs in our cohort. Notably, myositis had a remarkably high incidence rate, while gastrointestinal toxicities were not as common in our cohort, which is different from the setting of NSCLC where interstitial pneumonitis could be more prevalent [62]. The latter situation has been thought to be related to the cross-immunoreaction to autoantigens mediated with immunotherapy [21,62]. Interestingly, in our cohort, patients who received double immunotherapy were more likely to develop skin, gastrointestinal, and endocrine irAEs. Unlike the PD-1/PD-L1 axis, CTLA-4 blockade mediates the nonspecific expansion of naïve T cells [16]. Whether the application of anti-CTLA-4 antibodies in this study or a unique autoimmunity activation mechanism gives rise to this distinct irAE spectrum warrants further investigation.

The absolute or relative counts of peripheral blood cell components such as myeloid cells could reflect the magnitude of systemic inflammation, which plays an important role in tumorigenesis, cancer progression, and dissemination [24,63,64]. A high proportion of circulating neutrophils is correlated with an immunosuppressive tumor microenvironment and could denote poor prognosis in malignancies [63]. Lymphocytes, by contrast, are generally suppressors of tumorigenesis, and their expansion and infiltration in the tumor microenvironment are associated with potent immune responses [65]. Therefore, the equilibrium of circulating neutrophils and lymphocytes reflects the balance of protumor and antitumor forces. Dysregulation of the balance might be involved in the pathogenesis of irAEs. ECOG PS and baseline albumin reflect the physical status and nutrition reserves of patients, respectively, which have been widely recognized as prerequisites for antitumor responses [40]. LDH represents the metabolic activity of the body glucometabolic, which is proportional to tumor burden and is correlated with poor response to immunotherapy [66]. In summary, higher baseline albumin and lower NLR, ECOG PS, and LDH together depict a more competent immune status, predisposing patients to irAEs.

This study has several strengths. First, our study, to our knowledge, is the largest retrospective study investigating biomarkers of irAEs. Second, patients enrolled in our study exclusively received immunotherapy (excluding the concomitant application of target therapy, radiotherapy, or chemotherapy), which substantially eliminates the impact on peripheral blood components imposed by other medications and avoids the misdiagnosis of irAEs. Third, our study is the first to characterize the irAE spectrum of patients with gastrointestinal cancers who receive dual immunotherapy. However, our study also had limitations. First, this was a single-center study with a retrospective nature. Prospective validation is urgently needed to confirm the conclusions. Second, patients who were sensitive to immunotherapy (e.g., patients with MSI-H tumors) had a higher percentage in our cohort than in the real-world setting, probably leading to a selection bias. Third, other potential inflammatory indicators, such as C-reactive protein, were not included in the analysis due to incomplete data. Further investigations with regard to the utility of biomarker combinations are warranted.

## 5. Conclusions

In our current study, we demonstrated that early treatment lines, the presence of irAEs, and a lower C2 NLR were independent factors predicting superior ORR and PFS in patients with gastrointestinal cancers treated with immunotherapy. Lower ECOG PS, higher baseline albumin, and lower C2 NLR were independent risk factors for the development of irAEs. Patients who succumbed to irAEs during immunotherapy presented longer PFS and higher ORR. These findings could optimize the management strategy of patients with advanced gastrointestinal tumors treated with ICIs and facilitate the identification of patients who are more likely to respond to immunotherapy and suffer fewer toxicities. Prospective studies are warranted to verify these findings in clinical practice.

## Figures and Tables

**Figure 1 cancers-14-03736-f001:**
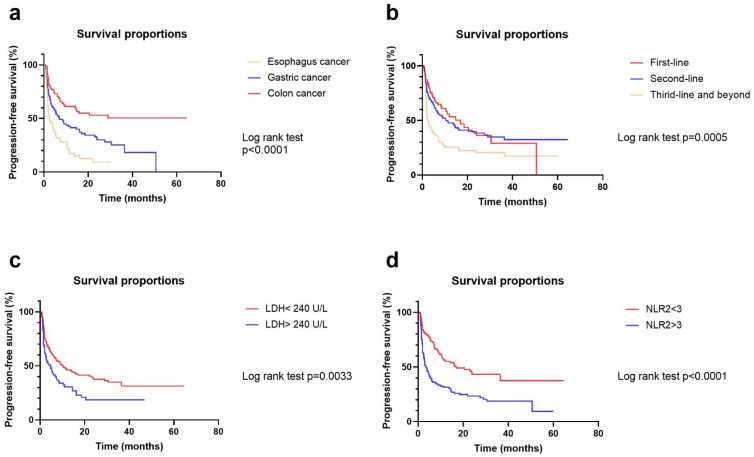
Survival curves according to tumor types (**a**), treatment lines (**b**), LDH (**c**), and NLR2 (**d**). Log-rank tests were used to evaluate survival differences.

**Figure 2 cancers-14-03736-f002:**
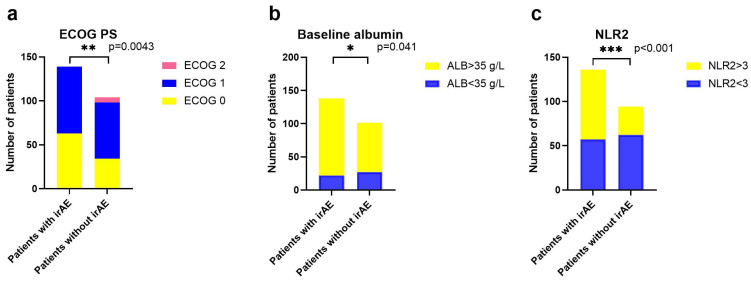
The development of irAEs corresponds to ECOG PS (**a**), baseline albumin (**b**), and NLR2 (**c**). Chi-square tests or Fisher exact tests were used to make intergroup comparisons. * *p* < 0.05, ** *p* < 0.01, *** *p* < 0.001 and not significant (ns).

**Figure 3 cancers-14-03736-f003:**
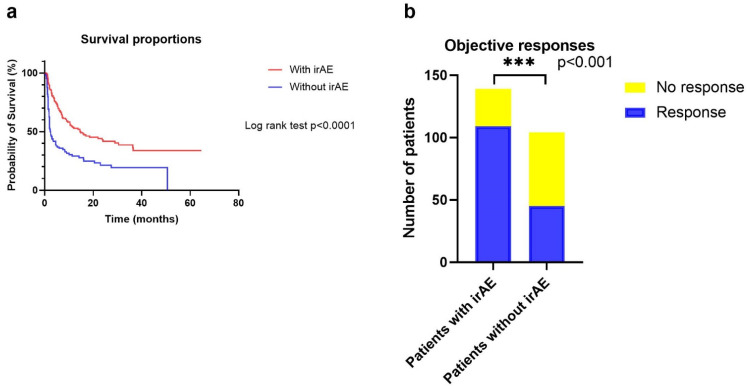
Correlations between irAEs and PFS (**a**) and ORR (**b**). Log-rank tests were used to evaluate survival differences. Chi-square tests or Fisher exact tests were used to make intergroup comparisons. *** *p* < 0.001.

**Table 1 cancers-14-03736-t001:** Demographic and clinicopathological characteristics of enrolled patients.

Variables	Number of Patients (N = 234)	Percentage (%)
Sex		
Female	71	29.2
Male	172	70.8
Age (years)		
<60	129	53.1
≥60	114	46.9
Primary tumor		
Esophagus cancer	50	20.6
Gastric cancer	115	47.3
Colon cancer	78	32.1
ECOG PS		
0	97	39.9
1	140	57.6
2	6	2.5
Treatment types		
Anti-PD-1 *	143	58.8
Anti-PD-L1 **	40	16.5
Anti-PD-1 + CTLA-4 ***	60	24.7
Line of immunotherapy		
First-line	70	28.8
Second-line	100	41.2
Third-line and beyond	73	30.0
PD-L1 expression		
Positive	99	40.7
Negative	38	15.7
Missing	106	43.6

ECOG PS, Eastern Cooperative Oncology Group performance status. * Regimens include: Nivolumab 3 mg/kg q2w n = 11; Pembrolizumab 200 mg q3w n = 25; Zimberelimab 240 mg q2w n = 24; Camrelizumab 200 mg q3w n = 5; Sintilimab 200 mg q3w n = 17; Tislelizumab 200 mg q3w n = 51; Toripalimab 3 mg/kg q2w n = 10. ** Regimens include: Atezolizumab 1200 mg q3w N = 4; Sugemalimab 1200 mg q3w n = 18; Envafolimab 10 mg/kg q3w n = 18. *** Regimens include: Nivolumab 1 mg/kg + Ipilimumab 3 mg/kg q3w n = 28; Cadolinimab 6 mg/kg q2w n = 32. q2w: every 2 week; q3w: every 3 week.

**Table 2 cancers-14-03736-t002:** Summary of immune-related adverse events.

Immune-Related Adverse Events (Categories)	Total Events (N = 260)	Immune-Related Adverse Events	Total Events (N = 260)	CTCAE Grade 1 (N = 170)	CTCAE Grade 2 (N = 58)	CTCAE Grade 3–4 (N = 32)
Skin	52	Pruritus	2	2	0	0
		Rash	50	34	12	4
Rheumatology	35	Arthralgia	8	6	2	0
		Myalgia	2	1	1	0
		Myositis/Elevated creatine kinase	20	8	3	9
		Dry mouth/Dry eye	1	1	0	0
		Dental ulcer	4	3	1	0
Pulmonary	9	Pneumonitis	9	5	4	0
Gastrointestinal	21	Nausea/Vomiting	4	3	1	0
		Diarrhea/Colitis	13	6	5	2
		Elevated amylase	4	2	1	1
Endocrine	49	Adrenocortical insufficiency	4	0	4	0
		Thyroiditis	42	39	3	0
		Hypophysitis	3	0	1	2
Hepatology	59	Transaminitis	32	21	5	6
		Hyperbilirubinemia	27	18	6	3
Cardiology	6	Myocarditis	3	0	1	2
		Arrhythmia	3	3	0	0
Nephrology	7	Proteinuria	6	2	3	1
		Elevated creatinine	1	0	0	1
Others	22	Fever	6	4	2	0
		Leukocytopenia	11	10	1	0
		Thrombocytopenia	2	0	2	0
		Dizziness/Headache	2	2	0	0
		Peripheral neuritis	1	0	1	0

CTCAE, common terminology criteria for adverse events.

**Table 3 cancers-14-03736-t003:** Univariate and multivariate analyses of ORR and PFS.

Variable	ORR						PFS						
	Univariate			Multivariate			References (HR = 1.000)	Univariate			Multivariate		
	OR	95% CI	*p* Value	OR	95% CI	*p* Value		HR	95% CI	*p* Value	HR	95% CI	*p* Value
Sex	0.585	0.321–1.067	0.080	0.442	0.205–0.954	**0.037**	Female	1.407	0.984–2.010	0.061			0.245
Age	1.000	0.981–1.019	0.995			0.794	<60 year	0.968	0.707–1.325	0.839	0.645	0.454–0.914	**0.014**
ECOG PS	0.701	0.433–1.135	0.149				0	1.268	0.916–1.756	0.152			
Tumor type	1.771	1.216–2.579	**0.003**			0.182	Colorectal	2.350	1.613–3.424	**<0.001**	2.628	1.792–4.014	**<0.001**
Treatment type	1.150	0.842–1.572	0.380				Monotherapy	0.809	0.551–1.188	0.280			
Treatment line	0.437	0.302–0.634	**<0.001**	0.484	0.310–0.754	**0.001**	≥3 line	0.534	0.385–0.739	**<0.001**	0.528	0.371–0.751	**<0.001**
PD-L1 expression	1.454	1.010–2.094	**0.044**			0.050	Positive	0.797	0.580–1.095	0.161			
Presence of irAE	4.393	2.519–7.660	**<0.001**	3.573	1.870–6.829	**<0.001**	No	0.494	0.360–0.677	**<0.001**	0.562	0.400–0.791	**0.001**
Highest CTCAE grade of irAE	1.767	1.345–2.320	**<0.001**			0.991	Grade 0–1	0.684	0.490–0.957	**0.026**			0.631
Baseline albumin	1.037	0.977–1.101	0.232				<35 g/L	0.645	0.302–1.379	0.258			
Baseline LDH	0.997	0.994–1.000	**0.021**			0.410	<240 U/L	1.642	1.171–2.300	**0.004**	1.563	1.088–2.247	**0.016**
Baseline hemoglobin	1.002	0.989–1.014	0.777				<90 g/L	0.833	0.448–1.552	0.565			
C1 NLR	0.972	0.900–1.050	0.468				<3	1.399	1.016–1.926	**0.040**			0.060
C1 PLR	1.000	0.998–1.002	0.858				<160	1.059	0.768–1.460	0.725			
C1 LMR	1.158	0.994–1.349	0.06			0.118	<3	0.603	0.440–0.827	**0.002**			0.711
C2 NLR	0.708	0.612–0.818	**<0.001**	0.737	0.629–0.864	**<0.001**	<3	2.108	1.517–2.928	**<0.001**	1.732	1.221–2.457	**0.002**
C2 PLR	0.997	0.994–0.999	**0.006**			0.411	<45	1.632	1.158–2.301	**0.005**			0.744
C2 LMR	1.618	1.322–1.981	1.618				<45	0.507	0.367–0.701	**<0.001**			0.162

ORR, overall response rate; PFS, progression-free survival; OR, odds ratio; CI, confidential interval; ECOG PS, Eastern Cooperative Oncology Group performance status; PD-L1, programmed cell death ligand-1; irAE, immune-related adverse event; CTCAE, Common Terminology Criteria for Adverse Event; LDH, lactate dehydrogenase; NLR, neutrophil to lymphocyte ratio; PLR, platelet to lymphocyte ratio; LMR, lymphocyte to monocyte ratio. *p* values were indicated in bold when statistical results were significant.

**Table 4 cancers-14-03736-t004:** Univariate and multivariate analyses of irAE.

Variable	Presence of irAE						Highest CTCAE Grade of irAE					
	Univariate			Multivariate			Univariate			Multivariate		
	OR	95% CI	*p* Value	OR	95% CI	*p* Value	OR	95% CI	*p* Value	OR	95% CI	*p* Value
Sex	0.587	0.330–1.044	0.070			0.056	0.660	0.374–1.165	0.152			0.151
Age	0.983	0.965–1.002	0.078			0.682	1.002	0.983–1.022	0.807			0.807
ECOG PS	0.514	0.317–0.835	**0.007**	0.571	0.334–0.976	**0.040**	0.645	0.394–1.055	0.081			0.079
Tumor type	1.486	1.036–2.131	**0.031**			0.293	1.332	0.918–1.934	0.131			
Treatment type	1.529	1.116–2.094	**0.008**			0.081	1.356	0.998–1.843	0.051			0.050
Treatment line	0.778	0.557–1.087	0.142				0.968	0.687–1.364	0.853			
PD–L1 expression	1.181	0.829–1.683	0.357				1.234	0.850–1.792	0.268			
Baseline albumin	1.076	1.013–1.142	**0.017**	1.067	1.000–1.138	**0.049**	1.022	0.961–1.087	0.486			
Baseline LDH	1.000	0.997–1.003	0.939				1.001	0.998–1.003	0.657			
Baseline hemoglobin	1.001	0.989–1.013	0.876				0.992	0.980–1.005	0.222			
C1 NLR	1.014	0.938–1.096	0.727				1.049	0.970–1.135	0.233			
C1 PLR	1.001	0.999–1.003	0.372				1.001	0.999–1.004	0.196			
C1 LMR	1.137	0.985–1.314	0.080			0.700	0.527	0.911–1.200	0.527			
C2 NLR	0.875	0.787–0.972	**0.013**	0.894	0.801–0.997	**0.044**	0.988	0.900–1.084	0.795			
C2 PLR	0.999	0.996–1.001	0.185				1.000	0.998–1.002	0.867			
C2 LMR	1.139	0.985–1.316	0.080			0.909	0.992	0.875–1.124	0.895			

ORR, overall response rate; PFS, progression-free survival; OR, odds ratio; CI, confidential interval; ECOG PS, Eastern Cooperative Oncology Group performance status; PD-L1, programmed cell death ligand-1; irAE, immune-related adverse event; CTCAE, Common Terminology Criteria for Adverse Event; LDH, lactate dehydrogenase; NLR, neutrophil to lymphocyte ratio; PLR, platelet to lymphocyte ratio; LMR, lymphocyte to monocyte ratio. *p* values were indicated in bold when statistical results were significant.

## Data Availability

The data presented in this study are available on request from the corresponding author.

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
