# Peer review of "Peripheral Blood Biomarkers Predictive of Efficacy Outcome and Immune-Related Adverse Events in Advanced Gastrointestinal Cancers Treated with Checkpoint Inhibitors"

_cancers, 2022, doi:10.3390/cancers14153736_

Round 1

Reviewer 1 Report

Summary:

Immune checkpoint inhibitor (ICI) therapy have revolutionized the treatment of some gastrointestinal cancers, such as microsatellite instable CRC and gastric and esophageal adenocarcinoma.

This is the largest retrospective study of peripheral blood biomarkers in patients with GI tumors treated with ICIs. Authors describe that ECOG, baseline albumin, and a low neutrophil-to-lymphocyte (NLR) ratio after 1 cycle of therapy (2-3 weeks after first ICI therapy) were independent factors for the development of immune-related adverse events (irAEs). They also show that the treatment line, the presence of irAEs, and a low NLR after one cycle of ICI therapy predicts a better ORR and PFS.

Major revisions:

-       In the ‘Results’ section immune-related adverse events (irAEs) are described per group of therapy, e.g. anti-PD-1, anti-PD-L1, or combination therapy. However, personally I don’t think this is detailed enough. Please add the specific therapies (e.g. nivolumab, pembrolizumab, ipilimumab plus nivolumab) to Table 1 and 2. For Table 2, please also describe the irAEs per regimen. Also, add the dosing in the legend, in order to interpret the data most optimally (I wonder for example which dose of ipilimumab is given? Was this 1 mg/kg or 3 mg/kg? Probably, 1 mg/kg since the irAEs of dual ICI treatment overall are quite low). Next, describe CTCAE grading in grade 1, grade 2, and grade 3-4 (in Table 2).

-       In Table 2 the diarrhea and enteritis are depicted separately. However, there are 5 patients with a grade 2-4 diarrhea, which is probable an immune-mediated colitis. I would suggest to take the two together and rename this into ‘Diarrhea/Colitis’.

-       In Figure 1B the survival proportions of first-line, second-line, and third-line and beyond are shown. However, in an extra sub figure, the survival proportions should also be shown per tumor type (for esophageal, gastric and colon cancer), in order to interpret this data well.

-       In Figure 1C the survival proportions are shown based on LDH. Here again, in an extra sub figure, the survival proportions based on LDH should also be shown per tumor type (for esophageal, gastric and colon cancer).

-       In Figure 1D the survival proportions are shown based on NLR2. Here again, in an extra sub figure, the survival proportions based on NLR2 should also be shown per tumor type (for esophageal, gastric and colon cancer).

-       In Figure 2 ECOG PS, albumin, and NLR2 seem to correspond with the occurrence of irAEs. However, a lot of addition data is lacking here. Was did seen in CRC, GC or esophageal cancer patients? And for NLR2, how were the results per treatment subgroup (anti-PD-1, anti-PD-L1, or combination therapy)? Please add this data in this Figure or in a Supplementary Figure.

-       The same applies for Figure 3. Please also show the results per tumor type and per treatment subgroup (anti-PD-1, anti-PD-L1, or combination therapy).

-       As discussed by the authors, this retrospective study population was enriched for MSI-high tumors. Therefore, the study results depicted in Figure 1-3 should also be shown for MSI vs. MSS tumors separately.

Minor comments:

-       Line 38 (introduction): please specify the revolutionized efficacy results in GI oncology, such as the successes in MSI-high CRC and gastric and esophageal adenocarcinoma. And also refer to the landmark papers here, such as the KEYNOTE-177 and CHECKMATE-649 studies.

-       Figure 2: please add the p-values of the asterisks to the legenda.

Author Response

Reviewer 1

Summary: Immune checkpoint inhibitor (ICI) therapy have revolutionized the treatment of some gastrointestinal cancers, such as microsatellite instable CRC and gastric and esophageal adenocarcinoma. This is the largest retrospective study of peripheral blood biomarkers in patients with GI tumors treated with ICIs. Authors describe that ECOG, baseline albumin, and a low neutrophil-to-lymphocyte (NLR) ratio after 1 cycle of therapy (2-3 weeks after first ICI therapy) were independent factors for the development of immune-related adverse events (irAEs). They also show that the treatment line, the presence of irAEs, and a low NLR after one cycle of ICI therapy predicts a better ORR and PFS.

Question 1: In the ‘Results’ section immune-related adverse events (irAEs) are described per group of therapy, e.g. anti-PD-1, anti-PD-L1, or combination therapy. However, personally I don’t think this is detailed enough. Please add the specific therapies (e.g. nivolumab, pembrolizumab, ipilimumab plus nivolumab) to Table 1 and 2. For Table 2, please also describe the irAEs per regimen. Additionally, add the dosing in the legend, in order to interpret the data most optimally (I wonder for example which dose of ipilimumab is given? Was this 1 mg/kg or 3 mg/kg? Probably, 1 mg/kg since the irAEs of dual ICI treatment overall are quite low). Next, describe CTCAE grading in grade 1, grade 2, and grade 3-4 (in Table 2).

Answer 1: We are grateful for your professional review of our manuscript. As you are concerned, the specific therapies are listed in the results section. We have made extensive adjustments to Table 1 and Table 2. Since the enrolled patients in our study came from the real world, the prescribed ICI drugs were diverse. Antibodies that target the same molecule have been manufactured by multiple companies. Therefore, statistical analyses of irAEs according to the brand name of ICIs could hardly be achieved due to limited sample size per product. Instead, we have supplemented the irAE data classified by different drug targets in Table S1. We consider this classification approach to make more sense and hope it might address your question.

Question 2: In Table 2 the diarrhea and enteritis are depicted separately. However, there are 5 patients with a grade 2-4 diarrhea, which is probable an immune-mediated colitis. I would suggest to take the two together and rename this into ‘Diarrhea/Colitis’.

Answer 2: We agree with your advice. Accordingly, we have revised the terms in Table 2.

Question 3: In Figure 1B the survival proportions of first-line, second-line, and third-line and beyond are shown. However, in an extra sub figure, the survival proportions should also be shown per tumor type (for esophageal, gastric and colon cancer), in order to interpret this data well.

Answer 3: Thank you for your kind suggestions. As suggested by the reviewer, we have added Figure S1 to show the results of the subgroup analysis. In summary, early treatment line is the positive prognostic factor for patients with esophageal cancer and colon cancer (see Figure S1a to S1c).

Question 4: In Figure 1C the survival proportions are shown based on LDH. Here again, in an extra sub figure, the survival proportions based on LDH should also be shown per tumor type (for esophageal, gastric and colon cancer).

Answer 4: Thank you for your kind suggestions. As suggested by the reviewer, we have added Figure S1 to show the results of the subgroup analysis. In summary, lower baseline LDH is a positive prognostic factor for patients with colon cancer (see Figure S1d to S1f).

Question 5: In Figure 1D the survival proportions are shown based on NLR2. Here again, in an extra sub figure, the survival proportions based on NLR2 should also be shown per tumor type (for esophageal, gastric and colon cancer).

Answer 5: Thank you for your kind suggestions. As suggested by the reviewer, we have added Figure S1 to show the results of the subgroup analyses. In summary, a lower C2 NLR is a positive prognostic factor for patients with gastric cancer and colon cancer (see Figure S1g to S1i).

Question 6: In Figure 2 ECOG PS, albumin, and NLR2 seem to correspond with the occurrence of irAEs. However, a lot of addition data is lacking here. Was did seen in CRC, GC or esophageal cancer patients? And for NLR2, how were the results per treatment subgroup (anti-PD-1, anti-PD-L1, or combination therapy)? Please add this data in this Figure or in a Supplementary Figure.

Answer 6: Thank you for your advice. Accordingly, we have added Figure S2 and Figure S3 to indicate the results of these subgroup analyses. In summary, gastric cancer patients with lower NLR2 and colon cancer patients with lower ECOG PS and/or lower NLR2 are more likely to develop irAEs when treated with ICIs. For patients treated with anti-PD-1 ± anti-CTLA-4 antibodies, those with a lower NLR2 bear a higher risk of developing irAEs.

Question 7: The same applies for Figure 3. Please also show the results per tumor type and per treatment subgroup (anti-PD-1, anti-PD-L1, or combination therapy).

Answer 7: Thank you for your advice. As you suggested, we have supplemented the corresponding contents in Figure S4.

Question 8: As discussed by the authors, this retrospective study population was enriched for MSI-high tumors. Therefore, the study results depicted in Figure 1-3 should also be shown for MSI vs. MSS tumors separately.

Answer 8: Thank you for your constructive comments. We have updated the microsatellite status information in the results section. It has been widely recognized that microsatellite status greatly influences the treatments as well as the prognosis of patients with gastrointestinal cancers, while it remains unclear whether it affects the incidence of irAEs. Since all esophageal cancers in our cohort were MSS tumors (MSI-H is a very rare event in esophageal cancers), we eliminated esophageal cancer and grouped gastric cancer and colon cancer together to perform the intergroup comparisons. Interestingly, in our cohort, there was no difference between MSS and MSI-H GI cancers regarding the incidence rate of irAEs. However, the values of ECOG PS, baseline albumin, and NLR2 in predicting irAEs were different in MSS versus MSI-H tumors (see Figure S5). These findings still need to be further verified in future studies.

Question 9: Line 38 (introduction): please specify the revolutionized efficacy results in GI oncology, such as the successes in MSI-high CRC and gastric and esophageal adenocarcinoma. And also refer to the landmark papers here, such as the KEYNOTE-177 and CHECKMATE-649 studies.

Answer 9: Thank you for your pertinent suggestions. We have added the efficacy results of several landmark studies to the introduction section.

Question 10: Figure 2: please add the p-values of the asterisks to the legenda.

Answer 10: Thank you for pointing this out. We have added the p-values to the figures accordingly.

Reviewer 2 Report

Through a retrospective longitudinal study, the present study aims to investigate the value of baseline peripheral blood markers for predicting treatment outcomes and IRAes among patients with gastrointestinal cancers treated with ICIs. 

In general, the study presents interesting evidence closely aligned with the various results in recently published clinical studies, especially concerning gastric cancer. One point clarified by the authors is that there is no description of the frequency of MSI-Status and TMB. Not counting on it dramatically reduces its relevance and novelty. However, it is possible to integrate a sensitivity analysis. A sensitivity analysis technique is developed to assess the sensitivity of interaction analyses to unmeasured confounding. Bias formulas for sensitivity analysis for interaction under unmeasured confounding are given on both additive and multiplicative scales. Simplified procedures are provided in case either one of the two factors does not interact with the unmeasured confounder in its effects on the outcome. An interesting consequence of the results is that if the two exposures of interest are independent (e.g. gene-environment independence), then even under unmeasured confounding, if the estimate of interaction is non-zero, then either there is a proper interaction between the two factors or there is an interaction between one of the factors and the unmeasured confounder; an exchange must be present in either scenario. The results are applied to two examples drawn from the literature. PMID: 21976358

Author Response

Reviewer 2

Through a retrospective longitudinal study, the present study aims to investigate the value of baseline peripheral blood markers for predicting treatment outcomes and IRAes among patients with gastrointestinal cancers treated with ICIs.

Question 1: In general, the study presents interesting evidence closely aligned with the various results in recently published clinical studies, especially concerning gastric cancer. One point clarified by the authors is that there is no description of the frequency of MSI-Status and TMB. Not counting on it dramatically reduces its relevance and novelty. However, it is possible to integrate a sensitivity analysis. A sensitivity analysis technique is developed to assess the sensitivity of interaction analyses to unmeasured confounding. Bias formulas for sensitivity analysis for interaction under unmeasured confounding are given on both additive and multiplicative scales. Simplified procedures are provided in case either one of the two factors does not interact with the unmeasured confounder in its effects on the outcome. An interesting consequence of the results is that if the two exposures of interest are independent (e.g. gene-environment independence), then even under unmeasured confounding, if the estimate of interaction is non-zero, then either there is a proper interaction between the two factors or there is an interaction between one of the factors and the unmeasured confounder; an exchange must be present in either scenario. The results are applied to two examples drawn from the literature. PMID: 21976358

Answer 1: Thank you for your advice. It is possible that there might be an unmeasured confounder associated with both peripheral blood constituents and the incidence of irAEs. Thus, we adopted your advice and conducted a sensitivity analysis for interactions under unmeasured confounding by utilizing E-values [ref. Ann Intern Med. doi:10.7326/M16-2607]. In our study, we found an association between NLR2 (dichotomous) and irAEs with an RR of 0.396 (CI, 0.231 to 0.680) (p=0.001). Then, according to the formula, E-value = 1/0.396 + √ {1/0.396 * (1/0.396-1)} = 4.49. E = 2.30 for the upper confidence limit. We could thus say that the observed risk ratio of 0.396 could be explained away by an unmeasured confounder that was associated with both the NLR2 and the irAEs by a risk ratio of 4.49-fold each, above and beyond the measured confounders, but weaker confounding could not do so; the confidence interval could be moved to include the null by an unmeasured confounder that was associated with both the NLR2 and the irAEs by a risk ratio of 2.30-fold each, above and beyond the measured confounders, but weaker confounding could not do so. Based on our clinical knowledge, an unmeasured confounder associated with the NLR2 and irAEs with a risk ratio of 4.49-fold seems implausible. The reviewer also pointed out that the microsatellite status might affect the incidence of irAEs. Accordingly, we have supplemented this information and have made further intergroup comparisons. Interestingly, in our cohort, there was no difference between MSS and MSI-H GI cancers regarding the incidence rate of irAEs (see Figure S5). Therefore, the evidence that the association between NLR2 and irAEs was causal arguably was strong, and we thought that a strong unmeasured confounder was unlikely to exist.

Reviewer 3 Report

The manuscript was aimed to define the predictive value of several blood parameters  in patients with gastrointestinal cancers treated with immune checkpoint inhibitors. In general, the manuscript is well written and presents  an interesting data which is in a proper fit with other recent publications as shown in  the reference list (the references 23-35).

The major concern is about the design of the study and diverse groups of patients enrolled in this study.

1) Indeed, only 28.8% of patients received immunotherapy as a first-line setting, all remaining patients received immunotherapy as a second or even third-line. Given that immune-related adverse effects might display the substantial delay (up to several months after the initiation of such therapy), it remains unclear how the authors addressed this issue. 

2) Do the authors observe any differences in the side-effects (frequency, severity, etc.)  between the groups of patients treated with different immune checkpoint inhibitors? If so, it should be also cleared in the manuscript and discussed. 

Author Response

Reviewer 3

The manuscript was aimed to define the predictive value of several blood parameters in patients with gastrointestinal cancers treated with immune checkpoint inhibitors. In general, the manuscript is well written and presents an interesting data which is in a proper fit with other recent publications as shown in the reference list (the references 23-35). The major concern is about the design of the study and diverse groups of patients enrolled in this study.

Question 1: Indeed, only 28.8% of patients received immunotherapy as a first-line setting, all remaining patients received immunotherapy as a second or even third-line. Given that immune-related adverse effects might display the substantial delay (up to several months after the initiation of such therapy), it remains unclear how the authors addressed this issue.

Answer 1: Thank you for your reasonable question. It is true that the time to onset of irAEs could be highly variable. However, according to the ESMO practice guideline [PMID: 28881921] and a large-scale meta-analysis of 23 clinical trials [PMID: 33171025], most irAEs occur within the first 3 months of commencing treatment. In that case, several previous studies use 12 weeks as a landmark for analyses [PMID: 35759816] [PMID: 35003896]. In our study, however, due to our concern of potential delayed irAEs, we extended the safety follow-up period to one year, which could cover the majority of irAE onsets. Relevant descriptions are further clarified in the methodology section.

Question 2: Do the authors observe any differences in the side-effects (frequency, severity, etc.)  between the groups of patients treated with different immune checkpoint inhibitors? If so, it should be also cleared in the manuscript and discussed.

Answer 2: Thank you for your pertinent suggestions. We have summarized the irAE data classified by different treatment types in Table S1 as you requested. In our cohort, we observed that combinational immunotherapy elicited a higher incidence of all-grade irAEs than anti-PD-1/PD-L1 monotherapy (73.3%, 54.5%, and 57.5%, respectively, p=0.044), while the proportions of grade ≥3 irAEs among the three groups of patients presented no significant difference (18.3%, 10.5%, and 17.5%, respectively, p=0.273). Specifically, we found that the incidence rates of irAEs affecting the skin, gastrointestinal system, and endocrine system were significantly higher in patients treated with anti-PD-1+anti-CTLA-4 agents, as suggested by chi-square analysis. These results indicate that patients who receive double immunotherapy are more likely to develop irAEs, which is consistent with multiple high-quality reports [PMID: 29320654] [PMID: 31053497]. Although it is not the main finding, we consider it appropriate to add the corresponding contents to the results and discussion sections.

Reviewer 4 Report

Dear authors,

In this manuscript authors examined the potentials of periphery blood biomarkers as a prognostic marker of ICI therapy and irAE in gastrointestinal cancers.

There are a few comments and changes authors should address:

1. in Line175, authors should double check the p-value in figure 2(figure2a p=0.04, which should be * instead of **) and maybe think a better way to present data in figure2. Current figure version is unclear what statistics and data (ratio not total number) the p-values are calculated based on.

2. in table 2, could authors clarify the last column CTCAE grade 2-4, not 3-4? authors discussed a lot about grade 3-4 and any grade, but not 2-4. I would suggest to add one more column to clarify.

3. author could write in a better and clear way when they include a few p-values in brackets, refer to line 151, 154, 160, 172, 175.

4. in discussion part, I would suggest authors do a detailed discussion on how these hematological index are implicated in other cancer types (line 229) if possible.

Author Response

Reviewer 4:

Dear authors, in this manuscript authors examined the potentials of periphery blood biomarkers as a prognostic marker of ICI therapy and irAE in gastrointestinal cancers. There are a few comments and changes authors should address:

Question 1: in Line175, authors should double check the p-value in figure 2(figure2a p=0.04, which should be * instead of **) and maybe think a better way to present data in figure2. Current figure version is unclear what statistics and data (ratio not total number) the p-values are calculated based on.

Answer 1: Thank you very much for your constructive suggestions. We need to clarify that all statistical analyses in this figure were performed by chi-square tests or Fisher’s exact tests (using total numbers but not ratios). Specific p values have been added to the figures, and the statistical tools have been stated in the figure legends.

Question 2: In table 2, could authors clarify the last column CTCAE grade 2-4, not 3-4? authors discussed a lot about grade 3-4 and any grade, but not 2-4. I would suggest to add one more column to clarify.

Answer 2: We agree with you that the classification criterion of irAEs in the original version of table 2 was inappropriate. Thus, we have reorganized this table according to your suggestions.

Question 3: Author could write in a better and clear way when they include a few p-values in brackets, refer to line 151, 154, 160, 172, 175.

Answer 3: Thank you for pointing out these defects. We have altered the wording in the corresponding positions.

Question 4: in discussion part, I would suggest authors do a detailed discussion on how these hematological indexes are implicated in other cancer types (line 229) if possible.

Answer 4: Thank you for your suggestions. A more adequate discussion has been made on this subject.

Round 2

Reviewer 1 Report

No further comments. 

Reviewer 4 Report

Hi, there is no more comments to address.